# Registered report: RAF inhibitors prime wild-type RAF to activate the MAPK pathway and enhance growth

**Ajay Bhargava[1], Steven Pelech[2], Ben Woodard[3], John Kerwin[3], Nimet Maherali[4], Reproducibility Project: Cancer Biology***

[1]Shakti BioResearch, Woodbridge, United States; [2]Kinexus Bioinformatics Corporation, Vancouver, Canada; [3]Biotechnology Research and Education Program, University of Maryland, College Park, United States; [4]Harvard Stem Cell Institute, Cambridge, United States

**Abstract** The Reproducibility Project: Cancer Biology seeks to address growing concerns about reproducibility in scientific research by conducting replications of selected experiments from a number of high-profile papers in the field of cancer biology. The papers, which were published between 2010 and 2012, were selected on the basis of citations and Altmetric scores (*Errington et al., 2014*). This Registered Report describes the proposed replication plan of key experiments from 'RAF inhibitors prime wild-type RAF to activate the MAPK pathway and enhance growth' by Hatzivassiliou and colleagues, published in Nature in 2010 (*Hatzivassiliou et al., 2010*). Hatzivassiliou and colleagues examined the paradoxical response of RAF-WT tumors to treatment with RAF inhibitors. The key experiments being replicated include Figure 1A, in which the original authors demonstrated that treatment of a subset of BRAF^WT tumor cell lines with RAF small molecule inhibitors resulted in an increase in cell viability, Figure 2B, which reported that RAF inhibitor activation of the MAPK pathway was dependent on CRAF but not BRAF, and Figure 4A, where the dimerization of BRAF and CRAF was modulated by the RAF inhibitor PLX4720, but not GDC-0879. The Reproducibility Project: Cancer Biology is a collaboration between the Center for Open Science and Science Exchange, and the results of the replications will be published by *eLife*.

*For correspondence: fraser@scienceexchange.com

Group author details:
Reproducibility Project: Cancer Biology See page 17

## Introduction

Mutations activating the H/K/N-RAS>B/C-RAF>MEK1/2>ERK1/2 signaling pathways are commonly found in many types of cancer, making members of this pathway promising drug targets. Several small molecule inhibitors have been created that target the three RAF isoforms. However, early observations using these drugs noted a surprising paradox; while drugs targeting CRAF were able to inhibit CRAF activity in vitro, they paradoxically activated the MEK1/2>ERK1/2 signaling modules in vivo. This activation was not due to direct activation of signaling components downstream of RAF (*Hall-Jackson et al., 1999a*; *1999b*).

Hatzivassiliou and colleagues found that RAF inhibitors, while effective in blocking signaling in BRAF^V600E mutant (MT) cancer cell lines, paradoxically increased cell proliferation in BRAF^WT cancer cell lines (*Hatzivassiliou et al., 2010*). Their findings were published along with two other reports demonstrating similar results (*Heidorn et al., 2010*; *Poulikakos et al., 2010*) and provided a key insight into the mechanism of paradoxical RAF activation in BRAF^WT cells, showing that it depended on drug-induced dimerization of wild-type (WT) RAF isoforms, specifically CRAF.

In Figure 1A, Hatzivassiliou and colleagues treated 19 cancer cell lines, comprising 4 BRAF^V600E mutant lines, 7 RAF/RAS-WT lines, and 8 KRAS-MT lines, with varying concentrations of two RAF

inhibitors and calculated the $IC_{50}$ value for each drug in each cell line. They found that, although cancer cell lines carrying the BRAF[V600E] mutation were susceptible to the RAF inhibitors, BRAF[WT] cell lines were not. This experiment will be replicated in Protocol 1.

To elucidate whether CRAF or BRAF contributed to MEK signaling in RAF-treated KRAS-mutant cells, the authors used inducible shRNA cell lines to test whether BRAF or CRAF were necessary for the activation of MEK1/2 in HCT116 cells, which are KRAS mutant. As reported in Figure 2B, silencing CRAF reversed MEK activation upon treatment with the RAF inhibitors GDC-0879 and PLX4720. This experiment will be replicated in Protocol 2.

To test whether inhibitor priming was mediated by the inhibitors' conformational effects on the RAF kinase domain, the authors assayed BRAF-CRAF heterodimerization through a series of immunoprecipitation assays coupled with or without RAF inhibitors. In Figure 4A, they reported that the CRAF kinase domain forms a stable complex with the BRAF kinase domain when inhibitors are not present. However, in the presence of the RAF inhibitor PLX4720, this CRAF-BRAF heterodimer kinase domain interaction is destabilized. In the presence of DGC-0879, the CRAF-BRAF interaction is stabilized. This experiment will be replicated in Protocol 3.

Hatzivassiliou's work was published along with two companion papers; Heidorn and colleagues showed that drugs targeting BRAF[V600E] caused dimerization with CRAF and pathway activation (*Heidorn et al., 2010*), while Poulikakos and colleagues observed that paradoxical RAF activation only occurred in the context of BRAF[WT] (*Poulikakos et al., 2010*). In a subsequent study, Poulikakos and colleagues showed that dimerization was a critical factor in allowing a variant of BRAF[V600E] to demonstrate enhanced activity as compared to BRAF[V600E] (*Poulikakos et al., 2011*). Work by Joseph and colleagues confirmed the findings of Hatzivassiliou and colleagues that BRAF[V600E] cell lines were sensitive to treatment with the BRAF inhibitor PLX4720, while RAS mutant/BRAF[WT] or RAS/RAF WT cell lines were not, and that MEK1/2>ERK1/2 signaling was activated in these BRAF[WT] lines (*Joseph et al., 2010*). Lee and colleagues assayed a panel of BRAF[V600E], NRAS mutant, or BRAF/NRAS WT cell lines by treating them with PLX4720. They observed that PLX4720 inhibited ERK signaling in BRAF[V600E] cells, but they did not observe paradoxical MEK1/2>ERK1/2 activation in BRAF[WT] lines. They attributed this effect to a lower percentage of serum in their culture conditions as compared to those used in previous studies. They then examined colony formation to assess drug effects on cell survival, and saw strong growth inhibition exclusively in BRAF[V600E] cells (*Lee et al., 2010*). Paradoxical activation was also observed in BRAF[WT] cells by Carnahan and colleagues and by Halaban and colleagues (*Carnahan et al., 2010*; *Halaban et al., 2010*). A year later, Kaplan and colleagues published corroborating evidence that PLX4720 paradoxically activated MEK1/2>ERK1/2 signaling in BRAF[WT] cells. They also confirmed that silencing of CRAF abrogated this activation of MEK1/2>ERK1/2 signaling (*Kaplan et al., 2011*).

## Materials and methods

Unless otherwise noted, all protocol information were derived from the original paper, references from the original paper, or information obtained directly from the authors. An asterisk (*) indicates data or information provided by the Reproducibility Project: Cancer Biology core team. A hashtag (#) indicates information provided by the replicating lab. All references to Figures refer to the original study.

### Protocol 1: Assessing cell viability of a panel of cancer cell lines treated with RAF and MEK inhibitors

This protocol describes the treatment of a panel of human cancer cell lines with or without mutations in BRAF or RAS with drugs targeting RAF and MEK and assessing cell viability. This experiment is a replication of Figure 1A.

#### Sampling

- This experiment will be repeated four times.
  - See Power calculations for details.
- Each experiment consists of three cohorts:
  - Cohort 1: Cell lines treated with a range of concentrations of PLX4720
    - 20 µM
    - 10 µM

- 5 µM
- 2.5 µM
- 1.25 µM
- 0.625 µM
- 0.313 µM
- 0.156 µM
- 0.078 µM
  - Cohort 2: Cell lines treated with a range of concentrations of GDC-0879
    - 20 µM
    - 10 µM
    - 5 µM
    - 2.5 µM
    - 1.25 µM
    - 0.625 µM
    - 0.313 µM
    - 0.156 µM
    - 0.078 µM
  - Cohort 3: Cell lines treated with a range of concentrations of PD0325901
    - 20 µM
    - 10 µM
    - 5 µM
    - 2.5 µM
    - 1.25 µM
    - 0.625 µM
    - 0.313 µM
    - 0.156 µM
    - 0.078 µM
- Each cohort consists of three cell lines:
  - A375 cells
    - $BRAF^{V600E}$ mutant
  - MeWo cells
    - $RAF^{WT}/RAS^{WT}$
  - HCT116
    - RAS mutant

## Materials and reagents

| Reagent | Type | Manufacturer | Catalog # | Comments |
|---|---|---|---|---|
| A375 | Cells | ATCC | CRL-1619 | DMEM + 10% FBS |
| MeWo | Cells | ATCC | HTB-65 | EMEM + 10% FBS |
| HCT116 | Cells | ATCC | CCL-247 | McCoy's 5a Medium Modified + 10% FBS |
| PLX4720 | Inhibitor | Symansis | SY-PLX4720 | |
| PD0325901 | Inhibitor | Symansis | SY-PD0325901 | |
| GDC-0879 | Inhibitor | Selleckchem | S1104 | Replaces Genentech and Array BioPharma source |
| Fluorescent plate reader | Equipment | BioTek FLx800 | | |
| 15-cm cell culture plates | Equipment | Corning | 430599 | Original unspecified |
| 96-well plates | Materials | Corning | 3903 | |
| DMEM | Medium | ATCC | 30–2002 | Original unspecified |
| EMEM | Medium | ATCC | 30–2003 | Original unspecified |
| McCoy's 5a medium modified | Medium | ATCC | 30–2007 | Original unspecified |
| Fetal bovine serum (FBS) | Reagent | ATCC | 30–2020 | Original unspecified |
| DMSO | Reagent | Fisher | D128–500 | Original unspecified |
| Cell Titer Glo kit | Reagent | Promega | G7570 | |

## Procedure

Notes:

- A375 cells are maintained in DMEM supplemented with 10% FBS.
- MeWo cells are maintained in EMEM supplemented with 10% FBS.
- HCT116 cells are maintained in McCoy's 5a Medium Modified supplemented with 10% FBS.
  - All cell lines are kept at 37°C/5% $CO_2$.
- All cell lines will be sent for STR profiling and mycoplasma testing

1. Expand cell lines as needed in 15-cm plates.
2. Determine range of detection of replicating lab's plate reader:
   a. Plate 500 – $1.6 \times 10^4$ A375, MeWo, and HCT116 cells in quadruplicate wells in a 96 well plate with 100 µl of appropriate medium. Incubate 5 days.
      i. Plate medium alone (no cells).
      ii. 500 cells/well
      iii. 1000 cells/well
      iv. 2000 cells/well
      v. 4000 cells/well
      vi. 8000 cells/well
      vii. 16,000 cells/well
   b. Five days later measure cell viability with the Cell Titer Glo kit according to manufacturer's instructions.
      i. Plot relative luminescence to cells/well.
      ii. Use seeding density for each cell line that give sub-confluency in 5 days and where the signal is still in the linear range at the end of the assay.
3. Seed cells, at density determined in Step 2 above, in quadruplicate wells (technical replicates) in 96-well plates and incubate overnight. Note: Information in Steps 2 through 3 is derived from Hoeflich and colleagues (*Hoeflich et al., 2009*).
   a. Include control wells that contain media but no cells to control for background luminescence.
   b. Also include wells with cells that will remain untreated (no DMSO).
4. The next day, treat cells with varying doses of each drug diluted in DMSO first and then add fresh media (to avoid excess DMSO toxicity, keep final DMSO percentage below 0.2%).
   a. Perform serial dilutions of drugs in DMSO in a 96-well plate.
      i. Drug doses (twofold dilution series, see Sampling section for concentrations)
      ii. Vehicle control (DMSO)
   b. Further dilute compounds in fresh growth media.
   c. Replace media in wells with new media containing appropriate drug concentrations.
   d. d. As controls, include wells of untreated cells and wells of cells treated only with vehicle.
5. Incubate cells for 4 days.
   a. Do not replace media during this incubation.
6. Measure cell viability with the Cell Titer Glo kit according to the manufacturer's instructions.
   a. Record luminescence.
   b. For each treated well, subtract average background luminescence calculated from media-only wells. The background luminescence defines 0% viability (baseline).
   c. Normalize luminescence to the average of the vehicle treated cells. The vehicle-treated cells define 100% viability (top).
   d. Fit data to a four-parameter curve, with the top and baseline held constant at 100% and 0% each, and calculate the absolute $EC_{50}$ value (where the curve crosses 50% viability) for each drug treatment of each cell line.
      i. Only report $EC_{50}$ values that can be accurately estimated. Otherwise report as >20 µM or <0.078 µM.
7. Repeat Steps 3–6 independently three additional times.

## Deliverables

- Data to be collected:
  - All raw luminescence values

- Luminescence values adjusted to compensate for background luminescence
- Luminescence values normalized to vehicle treated cells.
- $EC_{50}$ values for each cell line and each drug treatment (as seen in Figure 1A)

## Confirmatory analysis plan

- Statistical analysis of the replication data:
  - n/a
- Meta-analysis of original and replication attempt effect sizes:
  - The replication data will be presented as a mean with 95% confidence intervals and will include the original data point, calculated directly from the graph, as a single point on the same plot for comparison.

## Known differences from the original study

- All known differences are listed in the 'Materials and reagents' section above with the originally used item listed in the comments section. All differences have the same capabilities as the original and are not expected to alter the experimental design.
- While the original experiment examined nineteen cell lines, the replication will be restricted to three cell lines; A375, representing the BRAF$^{V600E}$ mutant lines, MeWo, representing the RAF$^{WT}$/RAS$^{WT}$ cell lines, and HCT116, representing the RAS mutant cell lines. HCT116 is used in a subsequent experiment described in Protocol 2.
- The replicating lab will plate the cells in a 96-well plate as opposed to a 384-well plate.

## Provisions for quality control

All data obtained from the experiment - raw data, data analysis, control data, and quality control data - will be made publicly available, either in the published manuscript or as an open access dataset available on the Open Science Framework (https://osf.io/0hezb/).

- STR profiling and mycoplasma detection results

## Protocol 2: Assessing CRAF and BRAF roles in drug-dependent activation of MEK

This protocol describes the treatment of HCT116 (KRAS-MT) cells expressing doxycycline-inducible shRNAs against *BRAF* and *CRAF*, and treatment with RAF inhibitors followed by Western blot examination of activation of MEK. This experiment is a replication of Figure 2B.

## Sampling

- This experiment will be repeated seven times for a final power of ≥80%.
  - The original data presented is qualitative (representative images). In order to determine an appropriate number of replicates to perform initially, we have estimated the sample sizes required based on a range of potential variance.
  - See Power calculations for details.
- The experiment consists of two cohorts:
  - Cohort 1: HCT116 cells with dox-inducible shRNA against BRAF
  - Cohort 2: HCT116 cells with dox-inducible shRNA against CRAF
- Each cohort will receive the following treatments:
  - No dox:
    - DMSO
    - 0.1 µM PLX4720
    - 1 µM PLX4720
    - 10 µM PLX4720
  - Dox:
    - DMSO
    - 0.1 µM PLX4720
    - 1 µM PLX4720

- 10 µM PLX4720
    - No dox:
        - DMSO
        - 0.1 µM GDC-0879
        - 1 µM GDC-0879
        - 10 µM GDC-0879
    - Dox:
        - DMSO
        - 0.1 µM GDC-0879
        - 1 µM GDC-0879
        - 0 µM GDC-0879
- Lysates from each treatment are probed for:
    - phospho-MEK 1/2
    - total MEK 1/2
    - BRAF
    - CRAF
    - Actin (additional loading control)

## Materials and reagents

| Reagent | Type | Manufacturer | Catalog # | Comments |
|---|---|---|---|---|
| HCT116 cells expressing doxycycline-inducible shRNA directed against *BRAF* | Cells | Provided by original authors | | |
| HCT116 cells expressing doxycycline-inducible shRNA directed against *CRAF* | Cells | Provided by original authors | | |
| Doxycycline | Drug | Alfa Aesar | J67043-AE | Original unspecified |
| PLX4720 RAF inhibitor | Drug | Symansis | SY-PLX4720 | |
| GDC-0879 | Inhibitor | Selleckchem | S1104 | Replaces Genentech and Array BioPharma source |
| DMSO | Reagent | Sigma | D2650-5X5ML | Original unspecified |
| McCoy's 5a Medium | Medium | ATCC | 30-2007 | |
| Fetal bovine serum | Reagent | Seradigm | 1400-500G | Original unspecified |
| 15-cm cell culture plates | Materials | Corning | 430599 | Original unspecified |
| Protease inhibitor mixture; Complete Mini | Reagent | Roche Applied Science | 04693159001 | |
| Phosphatase inhibitor mix | Reagent | Pierce | 78420 | |
| SDS-PAGE (4–20%) Tris-Glycine | Materials | BioRad | 456–1094 | Original unspecified |
| Nitrocellulose membrane | Materials | BioRad | 170–4158 | Original unspecified |
| ECL detection reagents | Reagents | Amersham | RPN2232 | |
| Rabbit phospho MEK (pMEK) 1/2 (Ser217/221) antibody | Antibody | Cell Signaling Technology | 9121 | 1:1000 |
| Rabbit MEK 1/2 antibody (clone 47E6) | Antibody | Cell Signaling Technology | 9126 | 1:1000 |
| Mouse anti-BRAF antibody | Antibody | Santa Cruz Biotechnology | sc-5284 | 1:1000 |
| Goat Anti-Mouse IgG-HRP | Antibody | Pierce | 31432 | 1:5,000–1:200,000 |
| Goat Anti-Rabbit IgG-HRP | Antibody | Pierce | 31460 | 1:5,000–1:200,000 |
| Mouse anti-CRAF antibody | Antibody | BD | 610151 | 1:1000 |
| Mouse anti-ß-actin (HRP conjugate) (clone 8H10D10) | Antibody | Cell Signaling Technology | 12262 | Not originally used |
| Kaleidoscope prestained standards protein ladder | Reagent | BioRad | 161–0324 | Original unspecified |
| 4–15% Mini-PROTEAN TGX Stain-Free SDS-PAGE gel | Materials | BioRad | 456–8085 | Original unspecified |
| Trans-Blot Turbo Mini Nitrocellulose transfer packs | Equipment | BioRad | 170–4158 | Original unspecified |

## Procedure

Notes:

- Some information derived from Hoeflich and colleagues (*Hoeflich et al., 2006*).

- HCT116 cells are maintained in McCoy's 5a Medium modified supplemented with 10% FBS at 37°C/5% $CO_2$.
- All cells will be sent for STR profiling and mycoplasma testing.

1. Determine if concentration of doxycycline to induce knockdown of BRAF and CRAF in HCT116 cells needs to be optimized. Before beginning experiment, perform protocol details as outlined below, without drug treatment (Step 4) and only analyzing expression of BRAF, CRAF, and actin (Step 8). Perform with at least 1 well per group (with and without dox treatment for each cell line).
   a. Normalize BRAF and CRAF to actin levels.
   b. If level of depletion is not similar to reported levels in Figure 2B, further optimize conditions, such as increasing concentration of dox.
   c. Once conditions of knockdown are optimized, use for all replicates of experimental procedure.
2. Seed [#]$5x10^4$ cells/well in 24-well plates for treatment.
   a. Seed 16 wells per cohort.
      i. 8 wells will be treated with dox.
      ii. 8 wells will remain untreated.
3. Induce shRNA expression of appropriate wells by treatment with 2 mg/ml dox for 3 days.
   a. This condition will be checked in Step 1 of this protocol and optimized if needed. Once optimized, use that condition for all replicates.
4. Treat appropriate cells with varying concentrations of PLX4720 or GDC-0879 for 1 hr.
   a. Treat cells with varying doses of each drug diluted in DMSO first and then fresh media added (to avoid excess DMSO toxicity, keep final DMSO percentage below 0.2%).
   b. In each set of 8 wells (dox treated and untreated), treat as follows:
      i. Well 1: DMSO
      ii. Well 2: 0.1 µM PLX4720
      iii. Well 3: 1 µM PLX4720
      iv. Well 4: 10 µM PLX4720
      v. Well 5: DMSO
      vi. Well 6: 0.1 µM GDC-0879
      vii. Well 7: 1 µM GDC-0879
      viii. Well 8: 10 µM GDC-0879
5. Lyse cells and harvest protein:
   a. Rinse cells in ice cold PBS.
   b. Lyse cells in ice cold lysis buffer: 0.5% NP40, 20 mM Tris pH7.5, 137 mM NaCl, 10% glycerol, 1 mM EDTA plus protease inhibitor mixture-complete mini, and phosphatase inhibitor mix.
   c. Spin lysate at 12,000xg for 5 min at 4°C.
      i. Transfer lysate to fresh tube after spinning.
   d. Quantify protein by the BCA method.
6. Separate proteins by SDS-PAGE:
   a. [#]Adjust sample to 1.5 µg/µL with 2X Lammeli Buffer/$H_2O$.
   b. [#]Boil sample for 5 min at >90°C.
      i. Load [#]10–20 µg of protein per lane on a [#]4–15% SDS-PAGE gel.
      ii. Run alongside a size marker ladder.
      iii. Transfer to nitrocellulose membrane using a [#]Trans-Blot Turbo Mini according to the manufacturer's instructions.
         1. [#]Run at 25 V, 1 A for 30 min.
         2. *Confirm protein transfer by Ponceau staining.
7. Block membrane in 5% non-fat dried milk in TBST (20 mM Tris pH 7.5, 136 mM NaCl, 0.1% Tween-20).
8. Incubate membrane at 4°C overnight with primary antibodies [#]diluted in 5% milk in TBST:
   a. Mouse anti-BRAF; 1:1000 dilution; 86 kDa
   b. Mouse anti-CRAF; 1:1000 dilution; 74 kDa
   c. Rabbit anti-pMEK 1/2; 1:1000 dilution; 45 kDa
   d. Rabbit anti-total MEK 1/2; 1:1000 dilution; 45 kDa
   e. Mouse anti-ß-Actin-HRP; 1:1000 dilution; 42 kDa
      i. Run one gel/membrane per antibody; do not strip and reprobe membranes for multiple antibodies.

 ii. Note: Actin serves as a loading control to ensure equal loading of lanes (additional).

9. #Wash membranes 3 x 5 min in TBST.

10. Incubate with HRP-conjugated secondary antibodies #diluted 1:20,000 in 5% milk in TBST for 1 hr at room temperature.

11. Visualize bands with ECL detection kit according to manufacturer's protocol.
    a. Quantify band intensity.
    b. For each drug and dose in each cell line (treated with or without dox), normalize pMEK values to total MEK values.

12. Repeat Steps 2–11 independently six additional times.

## Deliverables

- Data to be collected:
  - Images of whole gel, including ladder, of shRNA optimization (Step 1).
  - Images of whole gel, including ladder (compare to Figure 2B).
  - Quantification of band intensities; phospho-protein levels normalized to total protein levels.

## Confirmatory analysis plan

- Statistical analysis of the replication data:
  - Compare band intensities across all groups.
    - Four-way ANOVA (2 x 2 x 2 x 4 factorial) of the normalized pMEK values for each cell line (with or without dox), drug (PLX4720 or GDC-0879), and dose (0, 0.1, 1, and 10 µM) followed by:
      - Two-way interaction contrast of normalized pMEK values from BRAF and CRAF shRNA cell lines (with or without dox) across varying doses of GDC-0879 with the following Bonferroni corrected comparisons:
        - BRAF shRNA cell line with dox compared to without dox (across varying doses of GDC-0879)
        - CRAF shRNA cell line with dox compared to without dox (across varying doses of GDC-0879)
      - Two-way interaction contrast of normalized pMEK values from BRAF and CRAF shRNA cell lines (with or without dox) across varying doses of PLX4720 with the following Bonferroni corrected comparisons:
        - BRAF shRNA cell line with dox compared to without dox (across varying doses of PLX4720)
        - CRAF shRNA cell line with dox compared to without dox (across varying doses of PLX4720)
- Meta-analysis of original and replication attempt effect sizes:
  - The replication data will be presented as a mean with 95% confidence intervals and will include the original data point, calculated directly from the representative image, as a single point on the same plot for comparison.

## Known differences from the original study

- All known differences are listed in the 'Materials and reagents' section above with the originally used item listed in the comments section. All differences have the same capabilities as the original and are not expected to alter the experimental design.
- The replication attempt will use actin as an additional loading control not used in the original study.

## Provisions for quality control

All data obtained from the experiment - raw data, data analysis, control data, and quality control data - will be made publicly available, either in the published manuscript or as an open access dataset available on the Open Science Framework (https://osf.io/0hezb/).

- STR profiling and mycoplasma detection results.
- Induced shRNA knockdown conditions will be checked, and optimized if needed, prior to proceeding with the experiment.
- Image of Ponceau staining confirming protein transfer.
- Protein loading will be confirmed using actin.

## Protocol 3: Biochemical heterodimerization assay with recombinant RAF proteins in the presence or absence of RAF inhibitors

This protocol describes how to perform immunoprecipitation and Western blot analysis with recombinant CRAF and BRAF kinase domains in the presence or absence of the RAF inhibitors PLX4720 or GDC-0879. Wild-type BRAF and BRAF$^{V600E}$ kinase domains will be tested in the presence of wild-type CRAF. This experiment is a replication of Figure 4A.

## Sampling

- This experiment will be repeated six times for a final power of $\geq$80%.
  - The original data presented are qualitative (representative images). In order to determine an appropriate number of replicates to perform initially, we have estimated the sample sizes required based on a range of potential variance. We will also determine sample size post hoc.
  - See Power calculations for details.
- Each experiment consists of two cohorts:
  - Cohort 1: CRAF + BRAF$^{WT}$
  - Cohort 2: CRAF + BRAF$^{V600E}$
- Each cohort is incubated with CRAF and treated for 1 hr with:
  - DMSO
  - 10 µM of PLX4720
  - 10 µM GDC-0879
  - 1 mM AMP-PCP
- Include a sample of CRAF, without BRAF, treated with DMSO (negative control)
- Immunoprecipitate CRAF from each sample and probe for CRAF and BRAF.

## Materials and reagents

| Reagent | Type | Manufacturer | Catalog # | Comments |
|---|---|---|---|---|
| pFBHTc delta N 1-416 BRAF | Plasmid | MRC-PPU | DU586 | Made at Genentech and Array BioPharma. Communication with authors. |
| pFBHTc delta N 1-416 BRAF$^{V600E}$ | Plasmid | MRC-PPU | DU603 | |
| Recombinant glutathione S-transferase (GST)-CRAF (Y340D/Y341D) kinase domain | Protein | Invitrogen | PV3805 | Original catalog # not specified |
| Assay buffer | Chemical | Specific brand information will be left up to the discretion of the replicating lab and recorded later | | |
| DMSO | Chemical | | | |
| PLX4720 | Inhibitor | Symansis | SY-PLX4720 | |
| GDC-0879 | Inhibitor | Selleckchem | S1104 | Replaces Genentech and Array BioPharma source |
| Adenylylmethylenediphosphonate (AMP-PCP) | Chemical | Sigma | M7510 | Original catalog # not specified |
| Rabbit anti-GST | Antibody | Cell Signaling Technology | 2622 | |
| Protein A agarose beads | Chromatography | Millipore | IP02 | Original catalog # not specified |
| SDS-PAGE gel | Western materials | Prepared in replicating lab | | Original unspecified |
| Precision Plus Protein All Blue Standards | Reagent | BioRad | 161–0393 | Original unspecified |

*Continued on next page*

*Continued*

| Reagent | Type | Manufacturer | Catalog # | Comments |
|---------|------|--------------|-----------|----------|
| Ponceau stain | Reagent | Sigma | P7170 | Not originally included |
| Nitrocellulose membrane | Material | Pall Corporation | PN 66485 | Original unspecified |
| Mouse anti-CRAF (clone 53) | Antibody | BD Biosciences | 610151 | |
| Mouse anti-BRAF | Antibody | Sigma | WH0000673M1 | |
| Goat anti-mouse IgG-HRP | Antibody | Invitrogen Molecular Probes | A11029 | Replaces Alexa Fluor 488 goat anti-mouse IgG from Invitrogen, cat# A11029 |
| ECL Plus Reagent | Detection assay | Lumigen | PS-3 | Replaces TyphoonTM Scanner from Amersham Bioscience |
| Fluor-S Max Scanner | Instrument | BioRad | | |

## Procedure

1. Express 6His-BRAF$^{WT}$ and 6His-BRAF$^{V600E}$ kinase domains (417–766) from pFBHTc delta N 1-416 BRAF and pFBHTc delta N 1-416 BRAF$^{V600E}$ vectors, respectively, using baculovirus cells, following lab standard procedures. Affinity purify proteins using a nickel column using lab standard procedures.
2. Add 500 nM 6His-BRAF$^{WT}$ kinase domain or 6His-BRAF$^{V600E}$ kinase domain and 500 nM GST-CRAF kinase domain to assay buffer.
   a. Assay buffer: 25 mM HEPES, pH 7.4, 10 mM MgCl$_2$, 0.01% (v/v) Triton X-100, and 2 mM DTT.
   b. Four samples of GST-CRAF + BRAF$^{WT}$
   c. Four samples of GST-CRAF + BRAF$^{V600E}$
   d. One sample with GST-CRAF alone.
3. Incubate samples for 1 hr at room temperature in the presence of a fixed concentration of compound or vehicle:
   a. DMSO
   b. 10 µM of PLX4720
   c. 10 µM GDC-0879
   d. 1 mM AMP-PCP
      i. For all compounds, keep final DMSO concentration to 0.25%
4. Immunoprecipitate GST-CRAF proteins with rabbit anti-GST antibody and protein A agarose beads following manufacturer's instructions.
5. Separate proteins by SDS-PAGE:
   a. Boil samples in SDS-Sample Buffer at 100°C for 5 min.
   b. Load samples on a SDS-PAGE gel.
      i. Electrophorese all samples on gel alongside a size marker ladder.
      ii. Perform electrophoresis at 30 mA for 55 min, monitoring the dye front until it comes off the gel.
   c. Transfer to nitrocellulose membrane at 250 mA per gel for 1 hr.
      i. *Confirm protein transfer by Ponceau staining.
6. Block membrane in 5% non-fat dried milk in TBST (20 mM Tris-HCl pH 7.5, 136 mM NaCl, 0.1% Tween-20) as recommended by manufacturer.
7. Incubate membrane at 4°C overnight with antibodies against:
   a. Mouse anti-CRAF; 1:1000; 66 kDa
   b. Mouse anti-BRAF; 1:1000; 44 kDa
8. Incubate with secondary antibody in 1X TBS for 1 hr at room temperature as recommended by manufacturer.
   a. Rinse the membrane at least three times with TBST.
9. Detect secondary antibody and visualize bands with an imager.
   a. Quantify band intensity.
   b. For each sample, normalize BRAF IP values to GST-CRAF IP values.
   c. For each BRAF variant, normalize compound treatment to DMSO.
10. Repeat experiment independently five additional times.

## Deliverables

- Data to be collected:
  - Full gel images with ladder positions marked
  - Quantification of band intensities
    - Raw measurements as well as normalized band intensities
  - Graph of mean band intensities across replicates

## Confirmatory analysis plan

- Statistical analysis of the replication data:
  - Bonferonni corrected one-sample *t*-tests of normalized $BRAF^{WT}$ values (normalized to GST-CRAF and then DMSO) of the following conditions compared to 1 (DMSO):
    - PLX4720
    - GDC-0879
    - AMP-PCP
  - Bonferonni corrected one-sample *t*-tests of normalized $BRAF^{V600E}$ values (normalized to GST-CRAF and then DMSO) of the following conditions compared to 1 (DMSO):
    - PLX4720
    - GDC-0879
    - AMP-PCP
- Meta-analysis of original and replication attempt effect sizes:
  - The replication data will be presented as a mean with 95% confidence intervals and will include the original data point, calculated directly from the representative image, as a single point on the same plot for comparison.
- Additional exploratory analysis:
  - Two-way ANOVA of BRAF values (normalized to GST-CRAF) from DMSO, PLX4720, GDC-0879, or AMP-PCP treated samples for each BRAF variant incubated with GST-CRAF with the following Bonferroni corrected comparisons:
    - $BRAF^{WT}$ treated with DMSO compared to $BRAF^{WT}$ treated with PLX4720
    - $BRAF^{WT}$ treated with DMSO compared to $BRAF^{WT}$ treated with GDC-0879
    - $BRAF^{WT}$ treated with DMSO compared to $BRAF^{WT}$ treated with AMP-PCP
    - $BRAF^{V600E}$ treated with DMSO compared to $BRAF^{V600E}$ treated with PLX4720
    - $BRAF^{V600E}$ treated with DMSO compared to $BRAF^{V600E}$ treated with GDC-0879
    - $BRAF^{V600E}$ treated with DMSO compared to $BRAF^{V600E}$ treated with AMP-PCP

## Known differences from the original study

- All known differences are listed in the 'Materials and reagents' section above with the originally used item listed in the comments section. All differences have the same capabilities as the original and are not expected to alter the experimental design.
- The original data examined samples treated with AZ-628, a chemically unrelated ATP-competitive RAF inhibitor, which had a similar reported effect as GDC-0879. The replication will be restricted to examining only PLX4720 and GDC-0879, similar to the other experiments included in this replication attempt.
- The replicating lab will use a modified version of their in-house Western Blot protocol with antibodies analyzed by an ECL detection system instead fluorescence based.

## Provisions for quality control

All data obtained from the experiment - raw data, data analysis, control data, and quality control data - will be made publicly available, either in the published manuscript or as an open access dataset available on the Open Science Framework (https://osf.io/0hezb/).

- Image of Ponceau staining confirming protein transfer

## Power calculations

Note: for details on the full set of power calculations, see https://osf.io/a4e75/.

### Protocol 1

Summary of original data

- Values estimated from original published figure.

| Figure 1A: RAF inhibitors and MEK inhibitor | | EC50 (µM) |
|---|---|---|
| A375 | PLX4720 | 0.5 |
| | GDC-0879 | 0.3 |
| | PD0325901 | <0.0781 |
| MeWo | PLX4720 | >20 |
| | GDC-0879 | >20 |
| | PD0325901 | <0.0781 |
| HCT116 | PLX4720 | >20 |
| | GDC-0879 | >20 |
| | PD0325901 | 0.18 |

Power calculations

- Due to $EC_{50}$ values, such as PLX4720 and GDC-0879 with MeWo and HCT116 cells, unable to be determined (i.e. they are above 20 µM), this replication attempt will not compare values, but instead report $EC_{50}$ values that can be accurately estimated and compare them to the original reported values. If unable to obtain an $EC_{50}$ estimate the highest or lowest dose, depending on the situation, will be reported, similar to the original report.

### Protocol 2

Summary of original data reported in Figure 2B

- Summary of data:
  - Band densities were obtained with Image Studio Lite (LiCOR) from published images.
  - Normalization was performed by dividing the phospho-band intensity by the total protein band intensity.

Figure 2B; assumed number of biological replicates = 3

| hairpin | Dox | Drug | Dose | Normalized pMEK band density |
|---|---|---|---|---|
| shBRAF | - | GDC-0879 | 0 | 0.1453 |
| shBRAF | - | GDC-0879 | 0.1 | 0.8004 |
| shBRAF | - | GDC-0879 | 1.0 | 0.1263 |
| shBRAF | - | GDC-0879 | 10.0 | 0.0245 |
| shBRAF | + | GDC-0879 | 0 | 0.0457 |
| shBRAF | + | GDC-0879 | 0.1 | 0.6477 |
| shBRAF | + | GDC-0879 | 1.0 | 0.1485 |
| shBRAF | + | GDC-0879 | 10.0 | 0.0115 |
| shBRAF | - | PLX4720 | 0 | 0.0795 |
| shBRAF | - | PLX472 | 0.1 | 0.1615 |

*Continued on next page*

| shBRAF | - | PLX472 | 1.0 | 0.7760 |
| shBRAF | - | PLX472 | 10.0 | 1.0721 |
| shBRAF | + | PLX472 | 0 | 0.0861 |
| shBRAF | + | PLX472 | 0.1 | 0.1388 |
| shBRAF | + | PLX472 | 1.0 | 0.4522 |
| shBRAF | + | PLX472 | 10.0 | 0.9831 |
| shCRAF | - | GDC-0879 | 0 | 0.2121 |
| shCRAF | - | GDC-0879 | 0.1 | 0.7038 |
| shCRAF | - | GDC-0879 | 1.0 | 0.1554 |
| shCRAF | - | GDC-0879 | 10.0 | 0.0239 |
| shCRAF | + | GDC-0879 | 0 | 0.1005 |
| shCRAF | + | GDC-0879 | 0.1 | 0.1896 |
| shCRAF | + | GDC-0879 | 1.0 | 0.0933 |
| shCRAF | + | GDC-0879 | 10.0 | 0.0315 |
| shCRAF | - | PLX4720 | 0 | 0.4166 |
| shCRAF | - | PLX472 | 0.1 | 0.7469 |
| shCRAF | - | PLX472 | 1.0 | 1.1387 |
| shCRAF | - | PLX472 | 10.0 | 1.2976 |
| shCRAF | + | PLX472 | 0 | 0.2945 |
| shCRAF | + | PLX472 | 0.1 | 0.2449 |
| shCRAF | + | PLX472 | 1.0 | 0.2983 |
| shCRAF | + | PLX472 | 10.0 | 0.2729 |

The original data does not indicate the error associated with multiple biological replicates. To identify a suitable sample size, power calculations were performed using different levels of relative variance using the values quantified from the reported image as the mean. At each level of variance the effect size was estimated and used to calculate the needed sample size to achieve at least 80% power with the indicated alpha error. The achieved power is reported.

## Test family

- Four-way ANOVA: Fixed effects, special, main effects and interactions, alpha error = 0.05

## Power calculations

- Performed with G*Power software, version 3.1.7 (*Faul et al., 2007*).
- ANOVA F test statistic and partial $\eta^2$ performed with R software, version 3.2.2 (*R Core Team, 2015*).
  - For a given relative variance, 10,000 simulations were run and the F statistic and partial $\eta^2$ was calculated for each simulated data set.

| Groups | Variance estimate | F test statistic F(1,64) (shRNA, Dox, Drug interaction) | Partial $\eta^2$ | Effect size $f$ | A priori power | Total sample size (32 groups) |
|---|---|---|---|---|---|---|
| Normalized pMEK in shBRAF or shCRAF cells treated with or without Dox and varying doses of GDC-0879 or PLX4720 | 2% | 2513.61 | 0.97245 | 5.94063 | 99.9% | 96 |
| | 15% | 45.6115 | 0.39616 | 0.80999 | 99.9% | 96 |
| | 28% | 13.8318 | 0.16726 | 0.44817 | 99.1% | 96 |
| | 40% | 7.3203 | 0.09650 | 0.32681 | 88.4% | 96 |

## Test family

- ANOVA: Fixed effects, special, main effects and interactions, alpha error = 0.05 for two-way interaction contrast.

## Power calculations

- Performed with G*Power software, version 3.1.7 (*Faul et al., 2007*).

| Groups | Variance estimate | F test statistic F(1,64) (shRNA, Dox) | Partial $\eta^2$ | Effect size $f$ | A priori power | Total sample size (32 groups) |
|---|---|---|---|---|---|---|
| Normalized pMEK in shBRAF or shCRAF cells treated with or without Dox across varying doses of GDC-0879 | 2% | 327.18 | 0.83639 | 2.26101 | 99.9% | 96 |
| | 15% | 5.81651 | 0.08331 | 0.30147 | 82.9% | 96 |
| | 28% | 1.66928 | 0.02542 | 0.16150 | 82.1% | 320 |
| | 40% | 0.81795 | 0.01262 | 0.11305 | 81.5% | 640 |
| Normalized pMEK in shBRAF or shCRAF cells treated with or without Dox across varying doses of PLX4720 | 2% | 7267.4 | 0.99127 | 10.6561 | 99.9% | 96 |
| | 15% | 129.198 | 0.66873 | 1.42082 | 99.9% | 96 |
| | 28% | 37.0786 | 0.36683 | 0.76115 | 99.9% | 96 |
| | 40% | 18.1685 | 0.22111 | 0.53281 | 99.9% | 96 |

## Test family

- ANOVA: Fixed effects, special, main effects and interactions, Bonferroni's correction, alpha error = 0.025 for contrasts within each drug type

## Power calculations

- Performed with G*Power software, version 3.1.7 (*Faul et al., 2007*).

| Drug | Group 1 across dose | Group 2 across dose | Variance estimate | Effect size $f$ | A priori power | Samples per group |
|---|---|---|---|---|---|---|
| GDC-0879 | shBRAF (+ Dox) | shBRAF (- Dox) | 2% | 1.77874 | 99.9% | 3 |
| | | | 15% | 0.23717 | 84.6% | 6 |
| | | | 28% | 0.12705 | 81.2% | 19 |
| | | | 40% | 0.08894 | 80.4% | 38 |
| | shCRAF (+ Dox) | shCRAF (- Dox) | 2% | 4.97629 | 99.9% | 3 |
| | | | 15% | 0.66351 | 99.9% | 3 |
| | | | 28% | 0.35545 | 88.0% | 3 |
| | | | 40% | 0.24881 | 80.9% | 5 |
| PLX4720 | shBRAF (+ Dox) | shBRAF (- Dox) | 2% | 3.13815 | 99.9% | 3 |
| | | | 15% | 0.41841 | 96.2% | 3 |
| | | | 28% | 0.22415 | 86.2% | 7 |
| | | | 40% | 0.15690 | 82.9% | 13 |
| | shCRAF (+ Dox) | shCRAF (- Dox) | 2% | 18.2081 | 99.9% | 3 |
| | | | 15% | 2.42775 | 99.9% | 3 |
| | | | 28% | 1.30058 | 99.9% | 3 |
| | | | 40% | 0.91040 | 99.9% | 3 |

- Based on these power calculations, we will then run the experiment seven times. Each time we will quantify band intensity. We will determine the standard deviation of band intensity across the biological replicates and combine this with the reported value from the original study to simulate the original effect size. We will use this simulated effect size to determine the number of replicates necessary to reach a power of at least 80%. We will then perform additional replicates, if required, to ensure that the experiment has more than 80% power to detect the original effect.

## Protocol 3
## Summary of original data reported in Figure 4A

- Summary of data:
  - Normalization was performed by dividing the BRAF-band intensity by the captured GST-CRAF band intensity.

Figure 4A; assumed number of biological replicates = 3

| BRAF variant | Drug | Normalized BRAF band density |
|---|---|---|
| WT | DMSO | 1 |
| WT | AMP-PCP | 0.2 |
| WT | GDC-0879 | 2.7 |
| WT | PLX4720 | 0.2 |
| V600E | DMSO | 1 |
| V600E | AMP-PCP | 1.4 |
| V600E | GDC-0879 | 1.3 |
| V600E | PLX4720 | 1.3 |

The original data does not indicate the error associated with multiple biological replicates. To identify a suitable sample size, power calculations were performed using different levels of relative variance using the values quantified from the reported image as the mean. At each level of variance, the effect size was estimated and used to calculate the needed sample size to achieve at least 80% power with the indicated alpha error. The achieved power is reported.

## Test family

- Two-tailed $t$-test: Difference from constant (one sample case), Bonferroni's correction, alpha error = 0.00833.

## Power calculations

- Performed with G*Power software, version 3.1.7 (*Faul et al., 2007*).

| Group 1 (constant) | Group 2 | Variance estimate | Effect size $d$ | A priori power | Samples per group |
|---|---|---|---|---|---|
| WT BRAF (DMSO) | WT BRAF (AMP-PCP) | 2% | 200.00 | 99.9% | 3 |
| | | 15% | 26.667 | 99.9% | 3 |
| | | 28% | 14.286 | 99.4% | 3 |
| | | 40% | 10.000 | 91.8% | 3 |

*Continued on next page*

*Continued*

| Group 1 (constant) | Group 2 | Variance estimate | Effect size $d$ | A priori power | Samples per group |
|---|---|---|---|---|---|
| WT BRAF (DMSO) | WT BRAF (GDC-0879) | 2% | 31.482 | 99.9% | 3 |
| | | 15% | 4.1975 | 84.2% | 4 |
| | | 28% | 2.2487 | 82.0% | 6 |
| | | 40% | 1.5741 | 84.7% | 9 |
| WT BRAF (DMSO) | WT BRAF (PLX4720) | 2% | 200.00 | 99.9% | 3 |
| | | 15% | 26.667 | 99.9% | 3 |
| | | 28% | 14.286 | 99.4% | 3 |
| | | 40% | 10.000 | 91.8% | 3 |
| Sensitivity calculations | | | Detectable effect size $d$ | A prior power | Samples per group |
| BRAF$^{V600E}$ (DMSO) | BRAF$^{V600E}$ (AMP-PCP) | 2% | 8.0194 | 80.0% | 3 |
| | | 15% | 3.9581 | 80.0% | 4 |
| | | 28% | 2.1973 | 80.0% | 6 |
| | | 40% | 1.4917 | 80.0% | 9 |
| BRAF$^{V600E}$ (DMSO) | BRAF$^{V600E}$ (GDC-0879) | 2% | 8.0194 | 80.0% | 3 |
| | | 15% | 3.9581 | 80.0% | 4 |
| | | 28% | 2.1973 | 80.0% | 6 |
| | | 40% | 1.4917 | 80.0% | 9 |
| BRAF$^{V600E}$ (DMSO) | BRAF$^{V600E}$ (PLX4720) | 2% | 8.0194 | 80.0% | 3 |
| | | 15% | 3.9581 | 80.0% | 4 |
| | | 28% | 2.1973 | 80.0% | 6 |
| | | 40% | 1.4917 | 80.0% | 9 |

- Based on these power calculations, we will then run the experiment six times. Each time we will quantify band intensity. We will determine the standard deviation of band intensity across the biological replicates and combine this with the reported value from the original study to simulate the original effect size. We will use this simulated effect size to determine the number of replicates necessary to reach a power of at least 80%. We will then perform additional replicates, if required, to ensure that the experiment has more than 80% power to detect the original effect.

## Acknowledgements

The Reproducibility Project: Cancer Biology core team would like to thank the original authors, in particular Shiva Malek, for generously sharing critical information as well as reagents to ensure the fidelity and quality of this replication attempt. We thank Courtney Soderberg at the Center for Open Science for assistance with statistical analyses and Catherine Sutter at Kinexus Bioinformatics and Stephen Williams at the Center for Open Science for review and helpful comment on the manuscript. We would also like to thank the following companies for generously donating reagents to the Reproducibility Project: Cancer Biology; American Type Culture Collection (ATCC), Applied Biological Materials, BioLegend, Charles River Laboratories, Corning Incorporated, DDC Medical, EMD Millipore, Harlan Laboratories, LI-COR Biosciences, Mirus Bio, Novus Biologicals, Sigma-Aldrich, and System Biosciences (SBI).

# Additional information

## Group author details

Reproducibility Project: Cancer Biology

Elizabeth Iorns: Science Exchange, Palo Alto, United States; William Gunn: Mendeley, London, United Kingdom; Fraser Tan: Science Exchange, Palo Alto, United States; Joelle Lomax: Science Exchange, Palo Alto, United States; Nicole Perfito: Science Exchange, Palo Alto, United States; Timothy Errington: Center for Open Science, Charlottesville, United States

## Competing interests

AB: Shakti Bioresearch is a Science Exchange associated lab. BW, JK: Biotechnology Research and Education Program is a Science Exchange associated lab. RP:CB: EI, FT, JL, and NP are employed by and hold shares in Science Exchange Inc. The other authors declare that no competing interests exist.

## Funding

| Funder | Author |
| --- | --- |
| Laura and John Arnold Foundation | Reproducibility Project: Cancer Biology |

The Reproducibility Project: Cancer Biology is funded by the Laura and John Arnold Foundation, provided to the Center for Open Science in collaboration with Science Exchange. The funder had no role in study design or the decision to submit the work for publication.

## Author contributions

AB, SP, BW, JK, NM, Drafting or revising the article; RP:CB, Conception and design; Drafting or revising the article

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
