## [Decision Letter]

Thank you for submitting your work entitled "Registered Report: RAF inhibitors prime wild-type RAF to activate the MAPK pathway and enhance growth" for peer review at *eLife*. Your submission has been favorably evaluated by Sean Morrison (Senior editor) and five reviewers, one of whom is a member of our Board of Reviewing Editors.

The reviewers have discussed the reviews with one another and the Reviewing editor has drafted this decision to help you prepare a revised submission.

This is a proposal to replicate a study by Malek and co-workers that demonstrated a role for Raf inhibitors to regulate CRAF/BRAF dimerization, activate the MAPK pathway, and increase tumor growth. The conclusions of this study have been widely accepted in the field. Nevertheless, this replication study is worthwhile. Three experiments will be replicated: Figure 1A, 2A and 2B.

There are two major concerns with the proposed replication study:

1) The restriction of the analysis to the PLX drug in each study. The authors should examine both the GDC-0879 and PLX-4720 drugs because the binding modes of these molecules are different.

2) As noted in the Registered Report Introduction, paradoxical RAF activation was described previously. The advance provided by the Nature paper was to identify a mechanism. The proposed replication studies (Figure 1A, 2A and 2B) do not really address the mechanistic conclusions of the paper. Consequently, the registered report should be expanded, for example – to include data showing drug-induced CRAF/BRAF dimerization (Figure 4A).

Additional comments:

A) The authors state that GDC-0879 and PLX-4720 have similar effects; however, as described in this paper and other work, these 2 compounds have very different binding modes to BRAF based on the structural studies described by Hatzivassiliou et al. This results in different functional effects as shown in Figure 2C and Figures 4A-C. For instance, the PLX-4720 molecule exhibited significantly less CRAF activation compared to GDC-0879, exhibited far less BRAF/CRAF dimerization, and overall had a weaker effect on paradoxical activation compared to GDC-0879 (requiring much higher doses of drug). Based on these significant differences, both drugs should be used to replicate the study.

B) No information is provided as to how the cell viability dose response (EC_50_) data will be fit. The data presented in Figure 1A reports an absolute EC_50_ value and not a relative EC_50_. Also the highest dose of drug tested was 20 μM and compounds which are shown at 20 μM simply did not reach >50% inhibition at the highest dose tested (20 μM). This is apparent in Supplementary Figure 2 where the highest dose of GDC-0879 tested is 20 μM and little effect is seen at that dose. Hence it is more accurate to compare EC_50_ values in BRAF-WT cell lines selected to EC_50_>20 μM rather than the absolute 20 μM value listed in the subsection “Power calculations”. Also as described in point 1, GDC-0879 should also be used in these studies for appropriate comparisons.

C) The authors propose starting dosing PLX-4720 in the cellular viability studies at 160 μM; however, this compound is has decreased solubility in media or water. In order to conduct such a study, the authors need to first demonstrate that PLX-4720 is soluble in the media tested at this dose. The authors also do not provide information on whether the compound dilutions are conducted in DMSO first and then media is added. Due to solubility issues, the compound should first be serially diluted in DMSO and then media added, ensuring again the final DMSO concentration is not >0.2%.

D) The cellular viability experiments are being conducted in 96-well plates (16,000 cells per well) and not 384-well plates and the seeding densities selected (14,000 cells per well) are not optimized. The authors need to identify optimal seeding densities for each given cell line ensuring that over the 4-day period of time the cells are in a logarithmic growth phase. The growth rates of cell lines will vary depending on both the media used as well as differences in serum lots, incubator temperature and other effects.

E) For Protocols 2 and 3: Again both PLX-4720 and GDC-0879 should be included when replicating these data given the different mechanisms of these inhibitors. As can be seen in Figure 2B, the data with these 2 compounds are not identical. The outcome of this experiment is also highly dependent on the degree of BRAF or CRAF knockdown upon Dox treatment. In order to appropriately replicate the results, the authors first need to ensure that with 2 mg/ml dox after 3 days BRAF and CRAF is depleted to similar levels as that shown in Figure 2B. If they are not, the authors need to optimize conditions further to ensure that the degree of BRAF and CRAF depletion is similar. The stable sh-inducible knockdown cell lines are pools and not clonal and hence optimization of knockdown conditions will be required as growth conditions will vary in a different laboratory with different lots of doxycyclin, serum, etc. Moreover, total MEK should be utilized to normalize pMEK levels for data analysis. Protein loading should be confirmed using actin and not BRAF or CRAF protein levels, especially given the experiment involves depleting cells of BRAF and CRAF.

F) As the variance increases, the sample size calculated increases – for example, the estimated sample size increases from 11 to 13 to 17 when the variance increases from 15% to 28% to 40%. Where do the investigators plan to stop their assumed variance and select a sample size? Please clarify.

G) Just before Protocol 2 description, the authors say that they will perform additional replicates to ensure that the experiment has more than 80% power. This gives the impression that the authors will continue to add more samples until they reach statistical significance. Please clarify.

---

## [Author Response]

This is a proposal to replicate a study by Malek and co-workers that demonstrated a role for Raf inhibitors to regulate CRAF/BRAF dimerization, activate the MAPK pathway, and increase tumor growth. The conclusions of this study have been widely accepted in the field. Nevertheless, this replication study is worthwhile. Three experiments will be replicated: Figure 1A, 2A and 2B.

*There are two major concerns with the proposed replication study: 1) The restriction of the analysis to the PLX drug in each study. The authors should examine both the GDC-0879 and PLX-4720 drugs because the binding modes of these molecules are different.*We agree that both drugs should be added to the planned replication. The revised manuscript includes both compounds.

*2) As noted in the Registered Report Introduction, paradoxical RAF activation was described previously. The advance provided by the Nature paper was to identify a mechanism. The proposed replication studies (Figure 1A, 2A and 2B) do not really address the mechanistic conclusions of the paper. Consequently, the registered report should be expanded, for example* –

*to include data showing drug-induced CRAF/BRAF dimerization (Figure 4A).*We agree and have included Figure 4A in the revised manuscript. We are also removing Figure 2A from the replication attempt.

Additional comments: A) The authors state that GDC-0879 and PLX-4720 have similar effects; however, as described in this paper and other work, these 2 compounds have very different binding modes to BRAF based on the structural studies described by Hatzivassiliou et al. This results in different functional effects as shown in Figure 2C and Figures 4A-C. For instance, the PLX-4720 molecule exhibited significantly less CRAF activation compared to GDC-0879, exhibited far less BRAF/CRAF dimerization, and overall had a weaker effect on paradoxical activation compared to GDC-0879 (requiring much higher doses of drug). Based on these significant differences, both drugs should be used to replicate the study.

We agree that both drugs should be added to the planned replication. The revised manuscript includes both compounds.

*B) No information is provided as to how the cell viability dose response (EC_50_) data will be fit. The data presented in Figure 1A reports an absolute EC_50_ value and not a relative EC_50_. Also the highest dose of drug tested was 20 μM and compounds which are shown at 20 μM simply did not reach >50% inhibition at the highest dose tested (20 μM). This is apparent in Supplementary Figure 2 where the highest dose of GDC-0879 tested is 20 μM and little effect is seen at that dose. Hence it is more accurate to compare EC_50_ values in BRAF-WT cell lines selected to EC_50_>20 μM rather than the absolute 20 μM value listed in the subsection “Power calculations”. Also as described in point 1, GDC-0879 should also be used in these studies for appropriate comparisons.*Thank you for raising these important points. It was not clear from the original paper if the EC_50_ values were absolute or relative. Additionally, we have included how the EC_50_ value will be determined: specifically, vehicle treated cells define 100% viability and the media-only wells define 0% viability, with the absolute EC_50_ calculated as the concentration at which the curve crosses 50% viability when the top (100%) and baseline (0%) are held constant when fitting the four-parameter curve.

We agree that the values for PLX4720 and GDC-0879 are above 20 µM for RAF^WT^/RAS^WT^ and RAS^MUTANT^ cell lines, however the concentration at which they do reach 50% inhibition is unknown (or not possible). The initial approach was to use 20 µM as a conservative estimate of what that value might be. However, as it is unlikely to be reached without additional complications, as raised by the next comment (c) below, we are reframing the approach to perform the cellular viability experiments. We will perform the experiment exactly as originally reported. If a curve is unable to be fit, as was the case for the original report, the EC_50_ will be reported as >20 µM. Only EC_50_ values that can be accurately estimated will be reported and compared to the original reported values.

Finally, we have included the GDC-0879 compound in the revised manuscript.

*C) The authors propose starting dosing PLX-4720 in the cellular viability studies at 160 μM; however, this compound is has decreased solubility in media or water. In order to conduct such a study, the authors need to first demonstrate that PLX-4720 is soluble in the media tested at this dose. The authors also do not provide information on whether the compound dilutions are conducted in DMSO first and then media is added. Due to solubility issues, the compound should first be serially diluted in DMSO and then media added, ensuring again the final DMSO concentration is not >0.2%.*Thank you for raising this important point. First, we have removed any additional concentrations that were used in the original assay in the revised manuscript. As a result, this will potentially lead to reporting EC_50_ values as >20 µM, or <0.078 µM, similar to Hatzivasiliou et al., 2010. Only EC_50_ values that can be accurately estimated will be reported and compared to the original reported values. Second, we have clarified the methodology to reflect how the compound dilutions are conducted first in DMSO and then media is added as described in Hoeflich et al., 2009, which was referenced in Hatzivassiliou et al., 2010.

*D) The cellular viability experiments are being conducted in 96-well plates (16,000 cells per well) and not 384-well plates and the seeding densities selected (14,000 cells per well) are not optimized. The authors need to identify optimal seeding densities for each given cell line ensuring that over the 4-day period of time the cells are in a logarithmic growth phase. The growth rates of cell lines will vary depending on both the media used as well as differences in serum lots, incubator temperature and other effects.*We agree and have added an optimization protocol that will identify the seeding density of each cell to ensure the cells are in a logarithmic growth phase at the end of the time period. The results of the optimization protocol will define the seeding density used in the cellular viability experiments with the compounds.

E) For Protocols 2 and 3: Again both PLX-4720 and GDC-0879 should be included when replicating these data given the different mechanisms of these inhibitors. As can be seen in Figure 2B, the data with these 2 compounds are not identical. The outcome of this experiment is also highly dependent on the degree of BRAF or CRAF knockdown upon Dox treatment. In order to appropriately replicate the results, the authors first need to ensure that with 2 mg/ml dox after 3 days BRAF and CRAF is depleted to similar levels as that shown in Figure 2B. If they are not, the authors need to optimize conditions further to ensure that the degree of BRAF and CRAF depletion is similar. The stable sh-inducible knockdown cell lines are pools and not clonal and hence optimization of knockdown conditions will be required as growth conditions will vary in a different laboratory with different lots of doxycyclin, serum, etc. Moreover, total MEK should be utilized to normalize pMEK levels for data analysis. Protein loading should be confirmed using actin and not BRAF or CRAF protein levels, especially given the experiment involves depleting cells of BRAF and CRAF.

We agree that both drugs should be added to the planned replication. The revised manuscript includes both compounds. We also agree and have included an optimization step in the protocol for inducing knockdown of BRAF and CRAF by Dox treatment. A range of dox will be used to determine if 2 mg/ml dox is sufficient to knockdown BRAF and CRAF to levels similar to what was reported. If necessary, the concentration of dox will be adjusted prior to conducting the experiment.

We have pMEK levels normalized to total MEK, which is the DV used in the statistical analysis.

We agree that BRAF and CRAF are not ideal loading controls and have included actin as an additional measure not included in the original report.

*F) As the variance increases, the sample size calculated increases* –

*for example, the estimated sample size increases from 11 to 13 to 17 when the variance increases from 15% to 28% to 40%. Where do the investigators plan to stop their assumed variance and select a sample size? Please clarify.*At the beginning of each protocol the starting sample size is defined (for each group), as well as at the end of each power calculation section. The range of variance is used to provide some guidance about the anticipated scope of the effort. Because we do not know the variance in the originally reported result, or what variance the replication might obtain, this approach provides a way to select an appropriate starting point, and thus minimum sample size. As stated at the end of each power calculation we will then use the variation from the minimum sample to perform a power calculation in order to identify if more samples are needed. This is further explained in response to the question below.

G) Just before Protocol 2 description, the authors say that they will perform additional replicates to ensure that the experiment has more than 80% power. This gives the impression that the authors will continue to add more samples until they reach statistical significance. Please clarify.

In order to perform a proper power calculation to determine minimum sample size, both the difference between means and variance estimates are needed. Because we do not have the original observed variance for Figure 2B, we have performed power calculations on a range of potential effect sizes (using the originally reported ‘mean’ values and a range of variances to calculate Cohen’s *d*). This method acts as a guide for selecting a starting, minimum, sample size. After performing this minimum, pre-defined sample size we will use the variance from these replication samples together with the means from the original report to determine if additional biological replicates are needed. The additional samples will not be added until statistical significance is reached for the replication attempt, but rather additional samples will be added until the estimated original effect size (calculated using the originally reported ‘mean’ value and the replication variance) can be detected with at least 80% power.